# Learning from Positive and Unlabeled Data with Adversarial Training

## Abstract

Positive-unlabeled (PU) learning learns a binary classifier using only positive and unlabeled examples without labeled negative examples. This paper shows that the GAN (Generative Adversarial Networks) style of adversarial training is quite suitable for PU learning. GAN learns a generator to generate data (e.g., images) to fool a discriminator which tries to determine whether the generated data belong to a (positive) training class. PU learning is similar and can be naturally casted as trying to identify (not generate) likely positive data from the unlabeled set also to fool a discriminator that determines whether the identified likely positive data from the unlabeled set ($\mathcal{U}$) are indeed positive ($\mathcal{P}$). A direct adaptation of GAN for PU learning does not produce a strong classifier. This paper proposes a more effective method called *Predictive Adversarial Networks* (PAN) using a new objective function based on KL-divergence, which performs much better. Empirical evaluation using both image and text data shows the effectiveness of PAN.

## 1 Introduction

Positive-unlabeled (PU) learning learns a binary classifier from only Positive ($\mathcal{P}$) and Unlabeled ($\mathcal{U}$) examples with no labeled negative examples (Denis, 1998; Liu et al., 2002; Elkan & Noto, 2008; du Plessis et al., 2014; Bekker & Davis, 2018b). It has many applications in text analysis, bio-medicine, spam detection, recommendation, remote sensing, matrix and knowledge base completion, graph learning, etc (Li & Liu, 2003; Yu et al., 2004; Fusilier et al., 2015; Scott & Blanchard, 2009; Li et al., 2010; Cerulo et al., 2010; Calvo et al., 2007; Ren et al., 2015; Hsieh et al., 2015; Wu et al., 2017; Zupanc & Davis, 2018). This paper proposes an adversarial PU learning method inspired by GAN (generative adversary networks) (Goodfellow et al., 2014), as we found that the GAN framework naturally suits PU learning. GAN aims to generate data of a particular training class, which is like the positive class $\mathcal{P}$ in PU learning. GAN works by generating likely positive data using a generator $G(\cdot)$ to fool a discriminator $D(\cdot)$, which determines whether the generated data indeed belong to the training (positive) class. For PU learning, this is like choosing likely positive instances from the unlabeled set $\mathcal{U}$ (achieved by a classifier) also to fool a discriminator $D(\cdot)$. Thus, we can simply replace GAN's generator with a classifier $C(\cdot)$ to produce a PU learner.[1] Note that the classifier is also a discriminator, but we use the term *classifier* here to distinguish it from the original *discriminator* of GAN. Note also both $C(\cdot)$ and $D(\cdot)$ are neural networks.[2]

This paper first proposes a direct adaptation of GAN, called a-GAN (*adapted GAN*), for PU learning following the above idea (see Section 3). However, this simple method is unable to produce state-of-the-art results. It led us to propose a new objective function based on Kullback-Leibler (KL) divergence. Our final proposed method is called PAN (*Predictive Adversary Networks*) due to the use of the classifier to replace the generator in GAN along with a new adversarial training method. On the new objective function, it is clearly desirable to consider the overall performance of $C(\cdot)$ on the entire unlabeled set. We use a distance metric to measure whether $C(\cdot)$ can produce similar

---

[1]PU learning is analogous to GAN and vice versa. That is, if we could put all the data (e.g., images) that can be generated by GAN's generator in a set, the set should be regarded as unlabeled as it contains both good (positive) and bad (negative) images. Then what the generator does is like selecting good images to fool the discriminator, which is exactly what our classifier does with a set of *given* unlabeled data.

[2]Their exact architectures are given in the experiment section (see Training Details in Section 5.1) since $C(\cdot)$ and $D(\cdot)$ have different architectures for text and image.

predictions to those of $D(\cdot)$ for all the examples in $\mathcal{U}$. If $C(\cdot)$ gives similar predictions, it means that the examples obtaining high probabilities from $C(\cdot)$ also get high probabilities from $D(\cdot)$, achieving the goal of fooling $D(\cdot)$, which will give us a good final PU classifier. We employ KL-divergence as the distance metric to measure the difference between $C(\cdot)$ and $D(\cdot)$.

One important advantage of PAN is that it does not need the input of class prior probability, which many state-of-the-art systems need (see Section 2). In practice, the class prior is unknown (there are some existing methods to estimate it, see below). Hou et al. (2018) and Chiaroni et al. (2018) have employed GAN to generate positive and/or negative data and then use a separate learner (e.g., CNN) to learn the final PU classifier using the generated data, they are not adaptations of GAN like PAN and their generators generate only images. PAN can be applied to any data as it has no generator. We evaluate PAN using both text and image classification datasets and show it outperforms start-of-the-art PU learning baselines even when we give them the perfect class prior probabilities. We also show that when the class prior probability estimation is off, the results can be quite poor.

## 2 RELATED WORK

Research in PU learning started in early 2000s (Denis, 1998; Liu et al., 2002; Lee & Liu, 2003; Yu et al., 2004; Liu et al., 2003; Elkan & Noto, 2008). Denis (1998) proposed it under statistical query framework. Liu et al. (2002) studied sample complexity. Elkan & Noto (2008) showed that if ranking is the goal of PU learning rather than classification, then PU learning is equivalent to the conventional binary learning. Due to numerous applications, there has been a recent surge of interest in PU learning (du Plessis et al., 2014; Mordelet & Vert, 2014; du Plessis et al., 2015; Sechidis & Brown, 2015; Claesena et al., 2015; Chang et al., 2016; Niu et al., 2016a; Yi et al., 2017; Liu et al., 2017; Jian et al., 2017; Northcutt et al., 2017; Kiryo et al., 2017; Sakai et al., 2017; Hou et al., 2018; Xu et al., 2017; Gong et al., 2018; Bekker & Davis, 2018a; Chiaroni et al., 2018; Hsieh et al., 2018; Bao et al., 2018; Shi et al., 2018; Sansone et al., 2018; Kato et al., 2019).

Early approaches for PU learning use 2-steps. Step 1 finds some *highly probable negative examples* from the unlabeled set. Step 2 uses the positive set, highly probable negative set, and the remaining unlabeled set to build a classifier (Liu et al., 2002; Yu et al., 2004; Li & Liu, 2003). Liu et al. (2003); Shi et al. (2018) also regarded the unlabeled data to have noisy labels. Several techniques applied training data re-weighting via regularization as well (Lee & Liu, 2003; Elkan & Noto, 2008). Work in (du Plessis et al., 2014; 2015; Kiryo et al., 2017) avoided parameter tuning by using unbiased risk estimators. Kato et al. (2019) deals with sample selection bias.

None of these works explores the idea of adversarial learning as we do in PAN. Hou et al. (2018) used GAN to generate positive and negative examples and then used them to build the final classifier employing a separate classification model. Chiaroni et al. (2018) proposed a method to use GAN to generate negative training examples. Both papers are for image classification. Generating text and other forms of data using GAN is relatively more challenging. PAN does not generate data. It replaces the generator with a classifier to directly train the required PU classifier in the adversarial learning framework using positive and unlabeled data. Our formulation and objective function are also quite different from those of GAN. Moreover, Xu et al. (2017) and Gong et al. (2018) used traditional margin-based methods.

Our a-GAN method is also related to the studies of weighted adversarial nets (WAN) (Chen et al., 2018; Zhang et al., 2018) as a-GAN also weights the discriminator, but the proposed PAN differs significantly from WAN. That is because although WAN weights the discriminator but the adversarial training procedure is the same as the original GAN (similar to our a-GAN). Our a-GAN is also similar to the method DAN recently posted on arXiv (Liu et al., 2019) as we can see from their Eq. 5 and our Eq. 2. The training methods are slightly different. PAN takes a very different approach as we will see in the next two sections.

Other related works include leveraging biased negative examples (Hsieh et al., 2018; Sakai et al., 2017), studying the random assumption of PU learning (Bekker & Davis, 2018a), multi-instance PU learning (Bao et al., 2018), scalable PU learning (Sansone et al., 2018), etc. More details can be found in the survey (Bekker & Davis, 2018b).

A weakness with these recent state-of-the-art systems is that they need the class prior probability (du Plessis et al., 2015; Kiryo et al., 2017; Hou et al., 2018; Xu et al., 2017; Chiaroni et al., 2018;

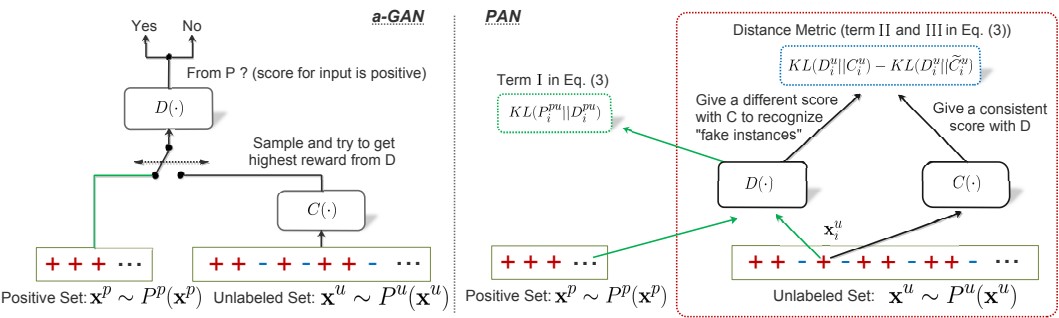

Figure 1: An Illustration of the objective functions of a-GAN (left) and PAN (right) as a comparison of the two models.

Kato et al., 2019), which is hard for the user to provide. Although there are methods that try to estimate the class prior (Menon et al., 2015; Ramaswamy et al., 2016; Jain et al., 2016; du Plessis et al., 2017; Gong et al., 2019), we will show that if the estimate is off, the results can be quite poor.

## 3 BACKGROUND

GAN is an *adversarial* learner that learns a generator for generating data such as images. It composes of two networks, a *generator* and a *discriminator*. The generator generates new data instances, and the discriminator evaluates them for authenticity, i.e. deciding whether each generated data instance it reviews belongs to the actual training dataset or not. Through an iterative and adversarial process, the generator can generate new data instances that the discriminator has hard time to distinguish from the real training data. GAN is formulated as a minmax game as follows:

$$\min_G \max_D \text{V}(D, G) = \mathbb{E}_{\mathbf{x} \sim P_{data}(\mathbf{x})}[\log D(\mathbf{x})] + \mathbb{E}_{\mathbf{z} \sim P_z(\mathbf{z})}[\log(1 - D(G(\mathbf{z})))] \quad (1)$$

where $G(\cdot)$ is the generator that aims to generate data that can approximate real data to fool the discriminator $D(\cdot)$, while $D(\cdot)$ tries its best to discriminate the generated data from real/training data. $P_{data}$ is the data generating distribution of the real data, and $P_z$ is the data generating distribution of the generator. Through an adversarial training, we expect to learn a good generator.

### 3.1 DIRECT ADAPTATION OF GAN FOR PU LEARNING

We now present the direct adaptation of GAN (a-GAN) for PU learning. The real training data in GAN is our labeled positive data $\mathcal{P}$. As illustrated in Figure 1, the discriminator $D(\cdot)$ in a-GAN still plays the same role as that in GAN, but the generator in a-GAN is replaced with another discriminator, which we call the *classifier* (or *predictor*) $C(\cdot)$. The goal of $C(\cdot)$ is to identify likely positives from the unlabeled set $\mathcal{U}$ to give to the discriminator for it to decide whether these are real positive data. The following equation shows this approach:

$$\min_C \max_D \text{V}(D, C) = \mathbb{E}_{\mathbf{x}^p \sim P^p(\mathbf{x}^p)}[\log D(\mathbf{x}^p)]$$
$$+ \mathbb{E}_{\mathbf{x}^s = \arg_{\mathbf{x}^u \sim P^u(\mathbf{x}^u)} C(\mathbf{x}^u)=1}[\log(1 - D(\mathbf{x}^s))] \quad (2)$$

where $\mathbf{x}_s$ denotes the example judged as a likely positive example from $\mathcal{U}$ by $C(\cdot)$. $P^p$ and $P^u$ are the data generating distributions of the known positive data and the unlabeled data in PU learning, respectively. The known positive examples are randomly selected from the positive population. The hidden positives in the unlabeled set is also a random sample of the same positive population.

Due to the discreteness of the last term in the equation, we use the Policy Gradient method (Sutton et al., 2000) from reinforcement learning to train it, where the last term is regarded as the reward for optimizing $C(\cdot)$. We call this adapted version of the system a-GAN (*adapted GAN*).

a-GAN does reasonably well (see Section 5.2). However, since it focuses on the positive data only, its learning is not balanced, which can cause some confusion with the separation of positive and negative data. Next, we present the final proposed method PAN, which emphasizes both positive and negative and is able to produce a better separation for them.

## 4 PROPOSED PAN

PAN adopts the same adversarial learning idea to build a PU classifier $C(\cdot)$. However, as the right part of Figure 1 shows, instead of using $D(\cdot)$ to directly discriminate the known positive data and the selected positive data by $C(\cdot)$, we propose to use the adversarial learning idea on the probability distributions of $D(\cdot)$ and $C(\cdot)$ on each example (or instance). Specifically, in the part surrounded by the red dash-lined box in Figure 1, $D(\cdot)$ and $C(\cdot)$ produce a score for each input example $\mathbf{x}_i^u$ from the unlabeled set in parallel with different optimization objectives. $D(\cdot)$ tries to give $\mathbf{x}_i^u$ the opposite prediction score to that of $C(\cdot)$ in order to identify it as a "fake" example; $C(\cdot)$ tries to give $\mathbf{x}_i^u$ a similar score to that of $D(\cdot)$ to fool $D(\cdot)$. The adversarial learning is performed through a distance metric, $D(\cdot)$ tries to enlarge the distance with $D(\cdot)$ but $C(\cdot)$ tries to shrink the distance, which is applied to each example in the unlabeled set (no sampling is used). We choose KL-divergence as the metric, which minimizes the information loss between the two probability distributions as it has been shown to be able to learn and suit complex distributions (Kingma & Welling, 2013; Goodfellow et al., 2014). The green links in Figure 1 show the procedure of optimizing the known positive data. Note that the unlabeled data are regarded as the negative data in PAN to endow $D(\cdot)$ the ability to recognizing negative samples to some extent.

### 4.1 PREDICTIVE ADVERSARY NETWORKS (PAN)

Unlike GAN, which only generates positive examples that are hard to distinguish by the discriminator $D(\cdot)$, we also want the remaining unlabeled examples to be easily distinguishable (as possible negatives) by the discriminator. To this end, $C(\cdot)$ tries to separate positive and negative examples in the unlabeled data with a large margin. That is, $C(\cdot)$ not only gives high probabilities to examples that $D(\cdot)$ has difficulty to distinguish (meaning $D(\cdot)$ also gives high probabilities to those examples) but also low probabilities to examples that are easy to distinguish by $D(\cdot)$ (meaning $D(\cdot)$ also gives low probabilities to those examples because of the easy separation). Note, when we say $C(\cdot)$ or $D(\cdot)$ gives high/low probability, we mean the probability of being positive. We propose to achieve our goal by controlling the distance (similarity) between the predictions of $C(\cdot)$ and $D(\cdot)$ on the unlabeled set. We use the sum of KL-divergences on the predictions of all examples (or instances) in $\mathcal{U}$ as the distance. In detail, PAN assumes the output of $D(\cdot)$ (respectively, $C(\cdot)$) on $i$th instance as a discrete distribution over binary outcomes (or classes) of *positive* and *negative*. For example, if $D(\cdot)$ (likewise, $C(\cdot)$) gives an instance the probability of 0.3. It means that for the positive outcome or class, the probability is 0.3, and for the negative outcome, the probability is 0.7. We use $D_i$ and $C_i$ to denote the two distributions and KL-divergence is employed to measure their distance. Superscripts $^{pu}$ and $^u$ denote the corresponding datasets. PAN's objective is defined as follows:

$$\min_C \max_D \ \mathrm{V}(D,C) = -\underbrace{\sum_{i=1}^{n} KL(P_i^{pu}||D_i^{pu})}_{\text{I}} + \lambda \big(\underbrace{\sum_{i=1}^{n_0} KL(D_i^u||C_i^u)}_{\text{II}} - \underbrace{\sum_{i=1}^{n_0} KL(D_i^u||\widetilde{C}_i^u)}_{\text{III}}\big) \tag{3}$$

where $P_i^{pu}$ is the probability distribution of positive and unlabeled of the $i$th instance (we basically treat *unlabeled* as negative, which is an issue to be addressed shortly) in the given PU data $X^{pu}$ (including both positive $X^p$ and unlabeled $X^u$ data), and $n$ and $n_0$ are the total numbers of training examples in $X^{pu}$ and $X^u$ respectively. $\widetilde{C}_i^u$ denotes the opposite distribution of $C_i^u$, i.e., $1 - C_i^u$ (with a slight abuse of notation). $\lambda$ is a hyper-parameter for balancing the distances.

We marked three terms in Eq. 3. The adversarial learning of Eq. 3 works as follows: The first term marked by I is to minimize the sum of the divergences between $D_i^{pu}$ and $P_i^{pu}$ (notice the minus sign in front). It aims to achieve the goal of helping $D(\cdot)$ recognize positive instances (it is necessary especially at the beginning of training). When optimizing $C(\cdot)$, the term marked by II minimizes the sum of the KL-divergences from $C_i^u$ to $D_i^u$, indicating that $C(\cdot)$ tries to give the same probability to the input $\mathbf{x}_i^u$ as $D(\cdot)$. In this case, the instances/examples getting high probabilities (chosen by $C(\cdot)$ as positive) can also get high probabilities from $D(\cdot)$. This achieves the goal of fooling $D(\cdot)$ by $C(\cdot)$. When optimizing $D(\cdot)$, the term marked II maximizes the sum of the KL-divergences between $D_i^u$ and $C_i^u$, meaning that $D(\cdot)$ tries to give low probabilities to the instances that get high probabilities from $C(\cdot)$ in order to detect 'fake' positive examples, and vice versa.

Using the first two terms can already perform the function of PAN. An advantage of PAN is that it can consider and optimize both positive and negative examples in the unlabeled set. However, we

show that the term marked II produces asymmetric gradient for positive and negative examples for both $D(\cdot)$ and $C(\cdot)$ in Appendix A. That means the term marked II can cause unbalanced training between positive and negative examples and lead to high precision and low recall. To this end, we propose the term marked III which can eliminate the concern (see Appendix A). Ablation study also shows the effectiveness of term III in Appendix D.3. With all three terms, we build an adversarial learning approach for PU learning through the minimizing and maximizing operations mentioned above, i.e., a minmax game between $D(\cdot)$ and $C(\cdot)$.

## 4.2 SIMPLIFICATION OF EQUATION 3

To facilitate the optimization of the objective function in Eq. 3, we use $D$ to denote $D(\mathbf{x}^{pu})$ and $C$ to denote $C(\mathbf{x}^{pu})$ and simplify Eq. 3 to (see *Appendix B* for derivations):

$$
\begin{aligned}
&\min_C \max_D \ \mathbf{V}(D, C) \\
&= \underbrace{\mathbb{E}_{\mathbf{x}^p \sim Pp(\mathbf{x}^p)}[\log D(\mathbf{x}^p)] + \mathbb{E}_{\mathbf{x}^u \sim Pu(\mathbf{x}^u)}[\log(1 - D(\mathbf{x}^u))]}_{\text{IV:} -H(P^L, D(\mathbf{x}^{pu}))} \\
&\quad + \lambda \cdot \mathbb{E}_{\mathbf{x}^u \sim Pu(\mathbf{x}^u)}[\underbrace{(\log(1 - C(\mathbf{x}^u)) - \log(C(\mathbf{x}^u)))}_{\text{V}}(2D(\mathbf{x}^u) - 1)] \\
&\hphantom{\quad + \lambda \cdot \mathbb{E}_{\mathbf{x}^u \sim Pu(\mathbf{x}^u)}[\underbrace{\qquad\qquad\qquad\qquad\qquad\qquad\qquad}_{\text{VI}}}
\end{aligned}
\tag{4}
$$

where $P^p$ denotes the distribution of the positive data. As we marked in Eq. 4, term IV is the cross entropy loss between $D(\mathbf{x}^{pu})$ and the ground-truth label distribution $P^{pu}$ of the PU data, denoted by $H(P^{pu}, D(\mathbf{x}^{pu}))$. About the term marked VI, we elaborate with the following two points:

**(1).** Term VI can be viewed approximately as a policy gradient reinforcement learning algorithm for training $C(\cdot)$ but with no sampling operation, if we regard $D(\mathbf{x}^u)$ as the reward and term V as the policy. Clearly, if $D(\cdot)$ outputs a high 'reward' that exceeds 0.5, meaning that $D(\cdot)$ judges the current input as a positive instance with high probability, Eq. 4 will maximize the probability of $C(\cdot)$ over the current input to fit the distribution of $D(\cdot)$. However, if $D(\cdot)$ outputs a low 'reward' below 0.5, minimizing Eq. 4 is equivalent to minimizing the probability of $C(\cdot)$ over the current input. As a consequence, the distribution of $C(\cdot)$ is made closer to $D(\cdot)$.

**(2).** The term marked with V in Eq. 4 is a comparison game between the likelihood $\log(C(\mathbf{x}^u))$ of choosing an example or the likelihood $\log(1 - C(\mathbf{x}^u))$ of not choosing an example $x^u$. If the choosing probability is greater than the not choosing probability, the value of term V is less than 0. Then to optimize $D(\cdot)$, maximizing Eq. 4 is equivalent to minimizing the probability of $D(\mathbf{x}^u)$. On the contrary, maximizing Eq. 4 is equivalent to maximizing the probability of $D(\mathbf{x}^u)$. Clearly, this is an adversarial learning method: for the case that term V is less than 0, $D(\cdot)$ tries to distinguish examples selected by $C(\cdot)$ (to give low probabilities to these examples). For the case that term VII is greater than 0, $D(\cdot)$ tries to give high probabilities to examples not selected by $C(\cdot)$, which helps the system move away from local training optimal.

**Analysis of the Learned Classifier** $(C(\cdot))$**:** Although the proposed PAN is quite different from original GAN, its running follows the adversarial procedure. The behaviors of optimal $C(\cdot)$ and $D(\cdot)$ are that $C(\cdot)$ gives the same prediction as $D(\cdot)$ while $D(\cdot)$ cannot move away from $C(\cdot)$, which means $D(\cdot)$ also give positive (or negative) scores to examples that get positive (or negative) scores from $C(\cdot)$.[3] The reason that $D(\cdot)$ cannot move away from $C(\cdot)$ is because moving away from $C(\cdot)$ will let $D(\cdot)$ make errors on known positive data. We give theoretical analysis of the optimal decision surface learned by PAN in Appendix C. Due to the complexity of PAN, we cannot give the precise optimal decision surface expression, but we show its properties in the Appendix C.

## 4.3 TRAINING ALGORITHM OF PAN

Algorithm 1 gives the training algorithm of PAN using stochastic gradient descent for conciseness. Note, our method is not limited to using stochastic gradient descent. In this work, we use Adam for optimization. The algorithm alternately trains the discriminator $D(\cdot)$ and the classifier $C(\cdot)$. In each step or iteration, the lines between 5 to 10 (not including 10) are for training $D(\cdot)$ and the lines after 10 are for training $C(\cdot)$. The details of the algorithm are self-explanatory.

---

[3]Positive (or negative) score means the score is greater (or lower) than 0.5.

---

**Algorithm 1** PAN training by minibatch stochastic gradient descent method.

---

**Input:** given positive training data $X^p$; given unlabeled training data $X^u$;
**Initial:** Randomly initialize $D(\cdot)$ and $C(\cdot)$;
**for** number of training steps **do**
    // Training $D(\cdot)$ $k$ steps, we set $k = 1$.
5:    **for** $k$ steps **do**
        • Sample a mini-batch of $m$ positive examples $\{\mathbf{x}_1^p, \ldots, \mathbf{x}_m^p\}$ from $X^p$;
        • Sample a mini-batch of $m$ unlabeled examples $\{\mathbf{x}_1^u, \ldots, \mathbf{x}_m^u\}$ from $X^u$;
        • Update $D(\cdot)$ by ascending its stochastic gradient:

$$\nabla_{\theta_d} \sum_{i=1}^m [\log D(\mathbf{x}_i^p) + log(1 - D(\mathbf{x}_i^u)) + \lambda(\log(1 - C(\mathbf{x}_i^u)) - \log C(\mathbf{x}_i^u))(2D(\mathbf{x}_i^u) - 1)]$$

    **end for**
10:    • Sample a mini-batch of $m$ unlabeled examples $\{\mathbf{x}_1^u, \ldots, \mathbf{x}_m^u\}$ from $X^u$;
    • Update $C(\cdot)$ by descending its stochastic gradient:

$$\nabla_{\theta_c} \sum_{i=1}^m [\lambda(\log(1 - C(\mathbf{x}_i^u)) - \log C(\mathbf{x}_i^u))(2D(\mathbf{x}_i^u) - 1)]$$

    **end for**

---

## 5 EXPERIMENTS

We now evaluate the proposed technique PAN and compare it with state-of-the-art baselines. Three text and two image classification datasets are used in our experiments.

(1). **YELP**: a collection of online reviews from Yelp. Each review is labeled with a star rating ranging from 1 to 5. The dataset is extracted from the Yelp Dataset Challenge 2015. (2). **RT**: a collection of online reviews from rotten tomatoes with sentiment labels *good* and *bad*. (3). **IMDB**: another collection of online review for binary sentiment classification. (4). **20News**: a collection of about 20,000 newsgroup documents, partitioned (nearly) evenly across 20 different news topics. (5). **MNIST**: a collection of 70,000 images of handwritten digits from 0 to 9. (6). **CIFAR10**: a collection of 60000 32x32 colour images of 10 classes, with 6000 images per class. See [4] for all datasets.

### 5.1 EXPERIMENT SETTINGS

**Data Preparation:** Since the five datasets are for traditional supervised learning with class labels, we need to prepare positive $\mathcal{P}$ and unlabeled $\mathcal{U}$ data for PU learning. We use two steps, after which we obtain the training and testing data for each dataset (on the left of the dataset name in Table 1).

Step 1 - *Constructing positive and negative data.* As not all datasets have 2 classes, we need to make each of them a two-class (positive and negative) dataset. RT and IMDB are already two-class datasets. For YELP, which has 5 classes, we remove the reviews with the class label of 3-stars, and split the remaining classes into two: one as the positive data (4 or 5 stars) and the other as the negative data (1 or 2 stars) (this is commonly done for sentiment classification (Pang & Lee, 2008)). Following the baseline (Kiryo et al., 2017), for 20News, topics 'alt.', 'comp.', 'misc.', and 'rec.' form the positive data, and topics 'sci.', 'soc.' and 'talk.' form the negative data. For MNIST, all images labeled with even numbers form the positive data and all images labeled with odd numbers form the negative data. For CIFAR10, classes airplane, automobile, ship and truck are used as the positive data and the rest as the negative data.

Step 2 - *Creating PU learning datasets.* After step 1, we get positive and negative training data for each dataset. We then build the PU learning training dataset, which includes positive and unlabeled data as follows. For each dataset (except CIFAR10), we randomly select 10% (5% for CIFAR10 for diversity). We also show more results by varying the ratio in Appendix D.1, the percent of positive

---

[4]See http://www.yelp.com/dataset_challenge for YELP, http://www.cs.cornell.edu/people/pabo/movie-review-data/ for RT, https://www.imdb.com/interfaces/ for IMDB, http://qwone.com/~jason/20Newsgroups/ for 20NEWS, http://yann.lecun.com/exdb/mnist/ for MNIST, http://www.cs.toronto.edu/ kriz/cifar-10-python.tar.gz for CIFAR10.

examples from the whole positive set as the known positive data $\mathcal{P}$ for PU learning. The unlabeled data $\mathcal{U}$ consists of the negative data and the remaining positive data in the dataset.[5]

**Baselines:** We use our a-GAN and five state-of-the art representative approaches as the baselines.

**a-GAN**. This is the direct adaptation of GAN given in Section 3.

**UPU** (du Plessis et al., 2015). This method proposed a general unbiased estimator that is also convex for loss functions meeting certain linear-odd conditions.

**NNPU** (Kiryo et al., 2017). This is a non-negative risk estimator for PU learning. When minimized, it is more robust against overfitting, and is able to use flexible models even given limited $\mathcal{P}$ data. Note that NNPU has two versions, the linear and the MLP versions. We give the results of the MLP version as it does better.

**NNPUSB** (Kato et al., 2019). This is a recent algorithm that extended NNPU with an additional mechanism for handling sample selection bias.

**GenPU** (Hou et al., 2018): This system uses the GAN framework and an array of generators and discriminators to generate both positive and negative data for PU learning.

**PMPU** (Gong et al., 2018): this is a traditional SVM based PU learning method.

It is important to note that both UPU and NNPU need the input of the class prior probability, which is often not available in practice. In our experiments, we provide them the correct class prior probabilities. Even with this favorable condition, they are still weaker than PAN, which does not need the class prior probability input. For UPU and NNPU, we use the opensource code from the authors and a third party[6], respectively. For NNPUSB, we use the original code provided by the authors. Note also we use the same network as these baselines, including architecture, number of parameters, and the optimization method. We also give them exactly the same positive and unlabeled data and the test data. For GenPU, we again use the code provided by the authors. For PGAN's results mentioned above, since there is no source code available, we used the best results reported in the paper.

**Training Details:** For fair comparison, we use the same classifier $C$ for all systems following the baselines. For text, a 2-layer convolutional network (CNN), with 5 * 100 and 3 * 100 convolutions for layers 1 and 2 respectively, and 100 filters for each layer, is used as the classifier $C(\cdot)$ and discriminator $D(\cdot)$. The word embeddings are also trained by the system. For MNIST, the classifier is a 3-layer MLP (with 2 hidden layers, more specifically, d-512-256-1) as it is fairly simple. The classifier for the CIFAR10 dataset was an all convolutional net: $(32 \times 32 \times 3)$-[C$(3 \times 3, 96)$] - C$(3 \times 3, 96, 2)$ - [C$(1 \times 1, 192)$] - C$(1 \times 1, 10)$ - 1000 - 1000 - 1, where each input is a $32 \times 32$ RGB image, C$(3\times3, 96)$ means 96 channels of $3\times3$ convolutions followed by ReLU, C$(3 \times 3, 96, 2)$ means a similar layer but with stride 2, etc. We set $\lambda$ in Eq. 3 and 4 to 0.0001, please see more details in Appendix D.2. We also balance the impact of positive and unlabeled data for term I in Eq. 3 in training; otherwise the positive examples will be dominated by the unlabeled data. We use 1:1 ratio of positive data and unlabeled data in each mini-batch in training. The network parameters are updated using the Adam algorithm with a learning rate of 0.0001. For baseline a-GAN, it needs pre-training of $D(\cdot)$. We use the original positive and unlabeled (regarded as negative) data to pre-train $D(\cdot)$ in order to enable its ability to classify positive and unlabeled data. We pre-train $D(\cdot)$ 3 epochs for the datasets.

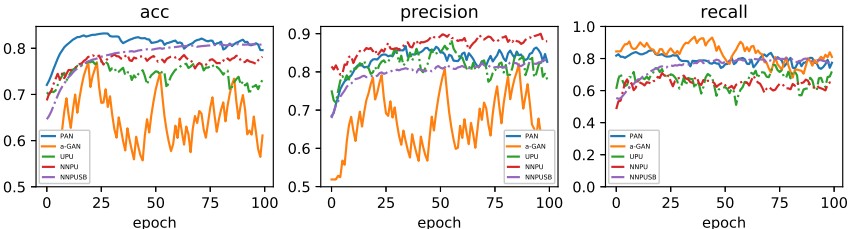

Figure 2: YELP - due to space limit, we only show 100 epochs.

---

[5]Note that PU learning has two data sampling settings, we use the one-pass (Niu et al., 2016b) or single-training-set (Elkan & Noto, 2008) setting, not the case-controlled or two-pass (Ward et al., 2009) setting.

[6]https://github.com/GarrettLee/nnpu_tf

Table 1: Dataset details and experiment results: On the left of Dataset - training and testing data for each dataset. On the right - F-score (F) and accuracy (Acc) of PAN and baselines for the dataset

| Training | | | Testing | | Dataset | a-GAN | | UPU | | NNPU | | NNPUSB | | PAN | |
|---|---|---|---|---|---|---|---|---|---|---|---|---|---|---|---|
| P-Label | Unlabel | | | | | | | | | | | | | | |
| Pos | Pos | Neg | Pos | Neg | | F | Acc | F | Acc | F | Acc | F | Acc | F | Acc |
| 26,000 | 234,000 | 260,000 | 20,000 | 20,000 | YELP | 83.72 | 83.33 | 79.72 | 79.33 | 80.70 | 81.06 | 81.92 | 81.76 | 83.45 | 83.56 |
| 426 | 3,839 | 4,264 | 1086 | 1047 | RT | 66.10 | 58.00 | 50.21 | 56.50 | 62.38 | 58.63 | 66.58 | 59.60 | 66.58 | 64.10 |
| 1,250 | 11,250 | 12,500 | 12,500 | 12,500 | IMDB | 73.01 | 70.64 | 70.35 | 69.87 | 76.21 | 74.62 | 74.24 | 71.88 | 77.10 | 78.84 |
| 800 | 7,144 | 6,056 | 1,800 | 1,800 | 20News | 63.48 | 68.66 | 59.13 | 53.07 | 78.52 | 78.07 | 75.87 | 75.56 | 81.06 | 81.00 |
| 3,000 | 29,492 | 30,508 | 4,926 | 5,074 | MNIST | 94.67 | 95.03 | 94.21 | 94.29 | 95.40 | 95.35 | 95.60 | 95.55 | 96.51 | 96.42 |
| 1,000 | 20,000 | 30,000 | 4,000 | 6,000 | CIFAR10 | 76.15 | 83.04 | 86.20 | 88.96 | 86.09 | 88.84 | 86.56 | 88.59 | 87.22 | 89.70 |
| - | - | - | - | - | Average | 76.24 | 76.45 | 73.30 | 73.67 | 80.34 | 79.43 | 80.13 | 78.82 | **81.99** | **82.27** |

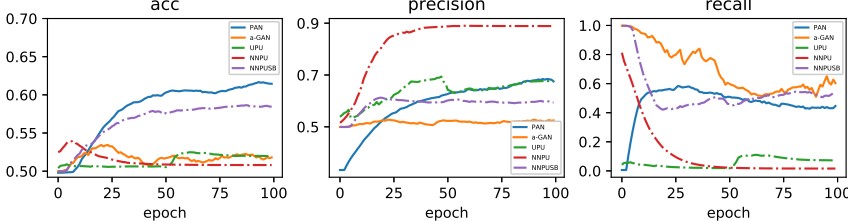

Figure 3: RT - due to space limit, we only show 100 epochs.

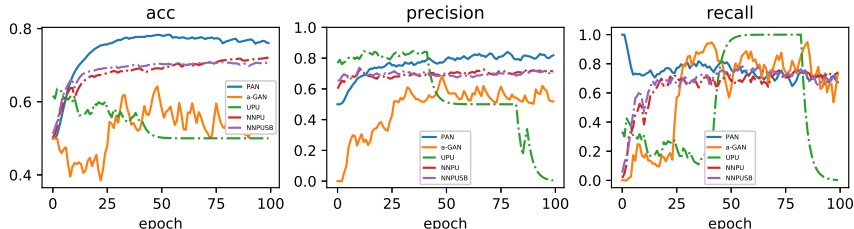

Figure 4: IMDB - due to space limit, we only show 100 epochs in this and the figures below.

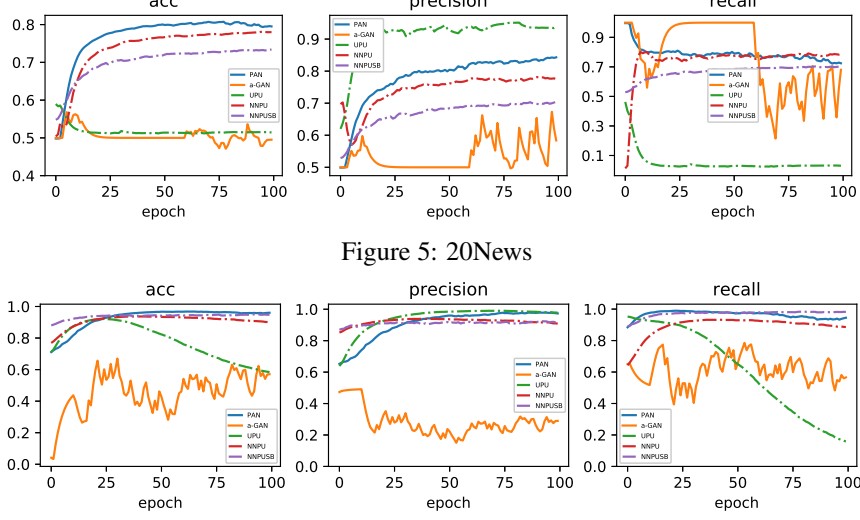

Figure 5: 20News

Figure 6: MNIST

## 5.2 RESULTS AND ANALYSIS

Figures 2-7 show the test accuracy, precision and recall in each epoch of each method. The usual first-order exponential weighted moving average smoothing with weight 0.7 is applied to the figures. Note that due to space limitations, we only show the curves of 100 epochs, but the final results in Table 1 are produced with more epochs as NNPU takes slightly longer to reach the peak (see below).

Figure 7: CIFAR10

The final accuracy and F-score results of A-GAN, UPU, NNPU, NNPUSB and PAN are given in Table 1 (on the right side of dataset names) and the results for GenPU and PMPU are given Table 2. Since different epochs give different results, for a fair comparison, we give the average of both the best F-score (F) and best accuracy (Acc) for each system on each dataset over 200 epochs (all systems converged before 200 epochs) over 5 runs. The F-score is measured on the positive data/class, as in PU learning the user is normally interested in identifying the positive data. The last row in the table gives the average result for each column.

From the Figures and Table 1, we can make the following observations.

(1). From the results in Table 1 (on the right side of dataset names in the table), we see that PAN outperforms all baselines on all datasets (the last row gives the average of each algorithm for all datasets). Among the baselines, NNPU and NNPUSB are the strongest and their results are very similar. PAN outperforms them markedly. Given that PAN does not need class prior probability input, this is even more significant. The direct adaptation of GAN a-GAN is weaker than both NNPU and PAN, but is better than UPU. Although NNPUSB extends NNPU, it did not do better than NNPU. The reason could be that our data do not have sample selection bias, which NNPUSB tries to address.

(2). Figures 2 to 7 show that PAN and NNPUSB are robust across all datasets with both high precision and high recall. And clearly PAN is better than NNPUSB. NNPU is rather unbalanced for precision and recall for YELP and RT, either very high precision but very low recall, or vice versa. a-GAN and UPU have the same problem, which is highly undesirable. PAN also outperforms baselines consistently in accuracy for all six datasets.

(3). From the 6 figures, we also see that a-GAN is unstable. Precision, recall, and accuracy fluctuate greatly from one epoch to another. Stability problem of GAN is well documented (Metz et al., 2017; Berthelot et al., 2017). Our adaptation requiring reinforcement learning to train it is likely to have made the problem worse.

(4). NNPU converges slowly (Figure 3). It didn't converge even at 100 epoch. We report its best accuracy and F-score in Table 1 in 200 epochs (it converged earlier than 200).

Table 2 shows the results of GenPU and PMPU. Since GenPU's data generator cannot generate text data and thus no results for the text datasets. We can see that GenPU's results are dramatically worse. Hou et al. (2018) showed that GenPU does well with few classes as the positive and negative, e.g, 1 class as positive and 1 class as negative. However, in our case, both positive and negative consists of many classes, which probably make GenPU work poorly. PMPU is also significantly poorer than PAN.

Table 2: Comparison between our method and GenPU and PMPU in terms of Accuracy (%)

| Dataset | GenPU | PMPU | PAN |
|---------|-------|-------|-------|
| MNIST | 70.43 | 95.74 | 96.46 |
| CIFAR10 | 66.25 | 81.34 | 89.65 |

In summary, we can conclude that PAN markedly outperforms the baselines in accuracy, F-score, robustness, and stability. Given that PAN does not need the class priors, this is more significant.

**Varying Positive Data and NNPU's Sensitivity to Class Prior:** We use MNIST and CIFAR10 as representatives to study these issues. For each dataset, we randomly select 1 or 2 classes in the original data to form the positive set, and the rest to form the negative set to generate 2 PU learning datasets as discussed above. Since both MNIST and CIFAR have 10 classes, for the 1-class positive PU data, the class prior probability is 10% for positive and 90% for negative (or 1:9 for short). For the 2-class positive PU data, it is 20% for positive and 80% for negative (or 2:8). The results are given in Table 2. We see similar improvements from PAN with the exact class prior given to NNPU

(1:9 or 2:8). For the 1:9 (respectively, 2:8) case for both MNIST and CIFAR10, if we change the class prior from the correct 1:9 (2:8) to the wrong 2:8 (3:7), NNPU's result drops are small (not in Table 2). So NNPU has some robustness. However, if we change to the wrong 3:7 or 4:6 (for the correct 1:9), and 4:6 or 5:5 (for the correct 2:8), the drops are dramatic for MNIST. For CIFAR10, they are smaller, even a small increase in F for the wrong 4:6 (correct 2:8), likely an anomaly as this data is hard, but still poorer than PAN. We conclude although it is possible to estimate the class prior (see Section 2), if the estimate is off, the results can be quite poor.

Table 3: Varying the positive data and the class prior probability for NNPU.

| Dataset | 1 class as positive (1:9) - results given as F / Acc | | | | 2 classes as positive (2:8) - results given as F / Acc | | | |
| | PAN | NNPU | | | PAN | NNPU | | |
| | | 1:9 (correct) | 3:7 (wrong) | 4:6 (wrong) | | 2:8 (correct) | 4:6 (wrong) | 5:5 (wrong) |
|---|---|---|---|---|---|---|---|---|
| MNIST | 97.88 / 99.19 | 97.82 / 99.16 | 91.16 / 96.29 | 82.13 / 91.10 | 95.59 / 98.42 | 95.55 / 98.32 | 77.50 / 88.90 | 66.94 / 81.41 |
| CIFAR10 | 51.94 / 84.73 | 51.46 / 84.65 | 48.55 / 82.79 | 41.99 / 76.64 | 59.42 / 78.35 | 56.45 / 78.27 | 57.67 / 77.61 | 54.62 / 73.29 |

## 6 CONCLUSIONS

This paper proposed a new GAN style PU learning method PAN based on adversarial training. PAN is also significantly different from GAN as PAN does not use a generator but a classifier in its place. The objective function of PAN is also entirely different as it is based on KL-divergence. PAN represents a new way to do PU learning. Empirical evaluation using both text and image datasets showed that PAN outperformed the state-of-the-art baselines. Also importantly, PAN obtained the better results without using any class prior probability information.

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

# Appendix

## A  ASYMMETRY OF $KL(D_i||C_i)$ FOR POSITIVE AND NEGATIVE DATA

In this section, we use the gradient asymmetry for positive and negative of KL-divergence to show the need for the term marked III in Eq. 3 (also see Eq. 8 below). The term marked II in Eq. 3 will produce asymmetric gradients for positive and negative examples for both $D(\cdot)$ and $C(\cdot)$ due to asymmetry of $KL(D_i||C_i)$ for positive and negative data (explained below). If we don't have the term marked III, gradient of $D(\cdot)$ is:

$$\nabla_D V(D,C) = \nabla_D[-\sum_{i=1}^{n} KL(P_i^{pu}||D_i^{pu}) + \lambda(\sum_{i=1}^{n_0} KL(D_i^u||C_i^u)]$$

$$= \underbrace{\sum_{i=1}^{n_p} \frac{1}{D(x_i^p)} - \sum_{i=1}^{n_u} \frac{1}{1-D(x_i^u)}}_{(a)} + \underbrace{\sum_{i=1}^{n_u} \log \frac{D(x_i^u)(1-C(x_i^u))}{(1-D(x_i^u))C(x_i^u)}}_{(b)} \tag{5}$$

where $n_p$ and $n_u$ are the size of positive and unlabeled set respectively. Term marked (a) is symmetric for positive and unlabeled data as they can obtain gradients with the same scale for the corresponding position, e.g., $D(x_i^p) + D(x_j^u) = 1$. But it is asymmetric for positive and negative data as positive data exist in the unlabeled set. That cause the positive being over optimized toward negative. Unfortunately, term marked (b) is also asymmetric for positive and negative data. We can see, the zero point of gradient term marked (b) is:

$$\log \frac{D(x_i^u)(1-C(x_i^u))}{(1-D(x_i^u))C(x_i^u)} = 0$$
$$\Rightarrow D(x_i^u) = C(x_i^u) \tag{6}$$

which means that the zero point is moved according to $C(x_i^u)$. In the worst case, if $C$ overfitted to give small probability to instances in the unlabeled set, then $D(\cdot)$ is not easy to escape from overfitting. In summary, that will cause high precision and low recall.

Asymmetric phenomenon also exist in Eq. 8 below without the term marked III as the gradient for $C(\cdot)$ is:

$$\nabla_C V(D,C) = \nabla_C[-\sum_{i=1}^{n} KL(P_i^{pu}||D_i^{pu}) + \lambda(\sum_{i=1}^{n_0} KL(D_i^u||C_i^u)]$$

$$= \underbrace{\sum_{i=1}^{n_u} \log \frac{C(x_i^u) - D(x_i^u)}{(1-C(x_i^u))C(x_i^u)}}_{(c)} \tag{7}$$

Clearly, it is asymmetric for positive and negative data, as positives have different gradient scale compared to negatives. And that can cause the unbalanced training problem. In this case, we propose to use the flipped distribution of $C_i^u$, denoted by $\widetilde{C}_i^u$, to address the problem, and please refer to the term marked III in Eq. 8. After adding the term marked III, the asymmetric gradient problem caused by the term marked II is eliminated. The gradient for $D(\cdot)$ now is $\sum_{i=1}^{n_u} \log \frac{(1-C(x_i^u))}{C(x_i^u)}$ which can be regarded as a constant when optimizing $D(\cdot)$. And the gradient for $C(\cdot)$ now is $\sum_{i=1}^{n_u} \log \frac{2D(x_i^u)-1}{(1-C(x_i^u))C(x_i^u)}$ which is symmetric between positive and negative.

## B  SIMPLIFICATION

Recall the loss function Eq. 3:

$$\min_C \max_D \ \mathrm{V}(D,C) = \underbrace{-\sum_{i=1}^{n} KL(P_i^{pu}||D_i^{pu})}_{\text{I}} + \lambda \overbrace{(\underbrace{\sum_{i=1}^{n_0} KL(D_i^u||C_i^u)}_{\text{II}} - \underbrace{\sum_{i=1}^{n_0} KL(D_i^u||\widetilde{C}_i^u))}_{\text{III}}}^{①} \tag{8}$$

KL-divergence is defined as:

$$KL(P||Q) = \sum_{x \in \mathcal{X}} P(x) \log P(x) - P(x) \log Q(x) \tag{9}$$

$\mathcal{X}$ denotes the probability space, it is 1 or 0 ($\mathcal{X} = \{1, 0\}$) in our scenario. We first address term I in Eq. 8, if we use $D$ to denote $D_i^{pu}(1)$ (the probability for $i$th instance being positive judged by discriminator) and $P$ to denote $P_i^{pu}(1)$, then $D_i^{pu}(0) = 1 - D$ and $P_i^{pu}(0) = 1 - P$. Then we have:

$$\begin{aligned}
&- KL(P_i^{pu}||D_i^{pu}) \\
&= -P \log P + P \log D - (1 - P) \log(1 - P) + (1 - P) \log(1 - D) \\
&= P \log D + (1 - P) \log(1 - D)
\end{aligned} \tag{10}$$

Due to the fact that $P_i^{pu}(0) = 0$ and $P_i^{pu}(1) = 1$ if the $i$th instance is positive and $P_i^{pu}(1) = 0$ and $P_i^{pu}(0) = 1$ if the $i$th instance is unlabeled, we can rewrite the result as:

$$\mathbb{E}_{\mathbf{x}^p \sim P^p(\mathbf{x}^p)}[\log D(\mathbf{x}^p)] + \mathbb{E}_{\mathbf{x}^u \sim P^u(\mathbf{x}^u)}[\log(1 - D(\mathbf{x}^u))] \tag{11}$$

Similarly, for term ① in Eq. 8, if we use $D$ to denote $D_i^u(1)$ and $C$ to denote $C_i^u(1)$, then we get:

$$\begin{aligned}
&KL(D_i^u||C_i^u) - KL(D_i^u||(1 - C_i^u)) \\
&= D \log D - D \log C + (1 - D) \log(1 - D) - (1 - D) \log(1 - C) - D \log D \\
&\quad + D \log(1 - C) - (1 - D) \log(1 - D) + (1 - D) \log C \\
&= -D \log C - (1 - D) \log(1 - C) + D \log(1 - C) + (1 - D) \log C \\
&= (\log(1 - C) - \log C)(2D - 1)
\end{aligned} \tag{12}$$

Then we have:

$$(\log(1 - C) - \log C)(2D - 1) = \mathbb{E}_{\mathbf{x}^u \sim P^u(\mathbf{x}^u)}[(\log(1 - C(\mathbf{x}^u)) - \log(C(\mathbf{x}^u)))(2D(\mathbf{x}^u) - 1)] \tag{13}$$

Combining Eqs. 11 and 13, we get Eq. 14:

$$\begin{aligned}
&\min_C \max_D \ V(D, C) \\
&= \underbrace{\mathbb{E}_{\mathbf{x}^p \sim P^p(\mathbf{x}^p)}[\log D(\mathbf{x}^p)] + \mathbb{E}_{\mathbf{x}^u \sim P^u(\mathbf{x}^u)}[\log(1 - D(\mathbf{x}^u))]}_{\text{IV}:\, -H(P^L, D(\mathbf{x}^{pu}))} \\
&\quad + \lambda \cdot \mathbb{E}_{\mathbf{x}^u \sim P^u(\mathbf{x}^u)} \underbrace{\underbrace{[(\log(1 - C(\mathbf{x}^u)) - \log(C(\mathbf{x}^u)))}_{\text{V}}(2D(\mathbf{x}^u) - 1)]}_{\text{VI}}
\end{aligned} \tag{14}$$

where $P^p$ denotes the distribution of the positive data.

## C  THEORETICAL ANALYSIS ABOUT THE LEARNED CLASSIFIER

In this section, we analyze the properties of the learned classifier and show why Eq. 3 can perform PU learning. Intuitively, from Eq. 8 (same as Eq. 3 in the paper), we can see that $D(\cdot)$ is biased if we only consider term I because the unlabeled set contains both positive and negative examples. However, terms II and III help correct the bias. Eq. 14 (same as Eq. 4 in the paper), which is derived from Eq. 8 for training, shows the property more clearly than Eq. 8. Notice that the bias in term I in Eq. 8 will result in high precision and low recall for the positive class. Now back to Eq. 14 and let us imagine that most examples in the unlabeled set are regarded as negative by $C(\cdot)$ (meaning low recall). From Eq. 14, we can see that the value of term V will be below zero. But when optimizing $D(\cdot)$, term VI will push $D(\cdot)$ up for these data points, and thus the bias is reduced and the low recall problem is mitigated because in the next optimization iteration, $C(\cdot)$ will follow $D(\cdot)$ to go up for these data points.

We now theoretically discuss the optimal decision surface of the classifier $C(\cdot)$ learned by the proposed PAN.

**Proposition 1.** Let $T(\mathbf{x}) = \log[1 - C(\mathbf{x})] - \log[C(\mathbf{x})]$, $\varepsilon(\mathbf{x}) = f(T(\mathbf{x}))$, the learned optimal decision surface of $C(\cdot)$ is:

$$\varepsilon(\mathbf{x}) = \frac{1}{2} - \frac{P^p(\mathbf{x})}{P^p(\mathbf{x}) + P^u(\mathbf{x})} \tag{15}$$

$f(\cdot)$ is a type of function that satisfies $\varepsilon(\mathbf{x}) \cdot T(x) > 0$.

From Proposition 1, we can see that PAN finds the decision surface by combining $P^p(\mathbf{x})$ and $P^u(\mathbf{x})$. The combination is controlled by $\varepsilon(\mathbf{x})$. $\varepsilon(\mathbf{x})$ is a function of $C(\mathbf{x})$.

## C.1 PROOF OF PROPOSITION 1

*Proof:* The training criterion for discriminator $D$, given any classifier $C$, is to maximize the quantity $V(C, D)$,

$$V(C, D) = \int_{\mathbf{x}} P^p(\mathbf{x}) \log D(\mathbf{x}) dx + \int_{\mathbf{x}} P^u(\mathbf{x}) \log(1 - D(\mathbf{x})) dx$$
$$+ \lambda \int_{\mathbf{x}} P^u(\mathbf{x})[\log(1 - C(\mathbf{x})) - \log C(\mathbf{x})](2D(\mathbf{x}) - 1) dx \tag{16}$$

Clearly, the maximum point appears at the point with derivative 0. Then we calculate the partial derivative of $V(C, D)$ to $D$ and get:

$$\underbrace{\frac{P^p(\mathbf{x})}{D(\mathbf{x})} - \frac{P^u(\mathbf{x})}{1 - D(\mathbf{x})}}_{\text{ⓐ}} + \underbrace{\lambda P^u(\mathbf{x})[\log(1 - C(\mathbf{x})) - \log C(\mathbf{x})]}_{\text{ⓑ}} = 0 \tag{17}$$

Directly computing the solution is complex. If we omit term ⓑ for the time being, the solution for Eq. 17 is $D(\mathbf{x}) = \frac{P^p(\mathbf{x})}{P^p(\mathbf{x})+P^u(\mathbf{x})}$. After bringing back ⓑ, this solution should be revised as follows. With the definition of $T(\mathbf{x}) = \log[1 - C(\mathbf{x})] - \log[C(\mathbf{x})]$ and $\varepsilon(\mathbf{x}) = f(T(\mathbf{x}))$, we can re-write the solution after revision:

$$D^*(\mathbf{x}) = \frac{P^p(\mathbf{x})}{P^p(\mathbf{x}) + P^u(\mathbf{x})} + \varepsilon(\mathbf{x}) \tag{18}$$

where $\varepsilon(\mathbf{x})$ is a function of $T(\mathbf{x})$ since $P^p(\mathbf{x})$ and $P^u(\mathbf{x})$ are decided by the dataset and $\varepsilon(\mathbf{x})$ changes with the change of $T(\mathbf{x})$. The exact expression of $\varepsilon(\mathbf{x})$ is difficult but we can show $\varepsilon(\mathbf{x}) \propto T(\mathbf{x})$ in our case. In Eq. 17, term ⓐ decreases monotonously when $D(\mathbf{x}) \in (0, 1)$,[7] and both $\lambda$ and $P^p(\mathbf{x})$ are greater than 0. In this case, if $T(\mathbf{x}) > 0$ ($T(\mathbf{x}) < 0$), to keep Eq. 17 equal to 0, $D(\mathbf{x})$ must move toward the positive (negative) direction, which indicates $\varepsilon(\mathbf{x}) > 0$ ($\varepsilon(\mathbf{x}) < 0$). Formally, we have:

$$\varepsilon(\mathbf{x}) \propto T(\mathbf{x}); \quad \varepsilon(\mathbf{x})T(\mathbf{x}) > 0 \tag{19}$$

Note that the training objective of $D$ can be interpreted as using the training data ($P^p(\mathbf{x})$ and $P^u(\mathbf{x})$) to find a discrimination bound and utilizing the learned knowledge in $C$ to adapt it. The mini-max game in Eq. 4 can now be reformulated as:

$$L(C) = \max_D V(C, D)$$
$$= \mathbb{E}_{\mathbf{x} \sim P^u(\mathbf{x})}[(\log(1 - C(\mathbf{x})) - \log(C(\mathbf{x})))(2D^*(\mathbf{x}) - 1)]$$
$$= \mathbb{E}_{\mathbf{x} \sim P^u(\mathbf{x}^u)}[T(\mathbf{x})(\frac{2P^p(\mathbf{x})}{P^p(\mathbf{x}) + P^u(\mathbf{x})} + 2\varepsilon(\mathbf{x}) - 1)] \tag{20}$$

Clearly, because the range of $T(\mathbf{x})$ is $(-\epsilon, \epsilon)$, $L(C)$ achieves its minimum when $T(\mathbf{x})$ and $(2D^*(\mathbf{x}) - 1)$ have opposite signs.[8] Then, the optimal $T^*(\mathbf{x})$ satisfies:

$$T^*(\mathbf{x})[\frac{2P^p(\mathbf{x})}{P^p(\mathbf{x}) + P^u(\mathbf{x})} + 2\varepsilon(\mathbf{x}) - 1] < 0 \tag{21}$$

As we introduced in Footnote 5, $C(\mathbf{x}) \in (0, 1)$. In this case, we use $C(\mathbf{x}) = 0.5$ as the decision surface to perform classification. Clearly, this decision surface equals to $T^*(\mathbf{x}) = \log(1 - 0.5) - \log(0.5) = 0$. In summary, we get the optimal decision surface:

$$\varepsilon(\mathbf{x}) = \frac{1}{2} - \frac{P^p(\mathbf{x})}{P^p(\mathbf{x}) + P^u(\mathbf{x})} \tag{22}$$

---

[7]We force $D(\mathbf{x})$ to satisfy the condition by adding a Sigmoid function to the end of $D$. We also force the output range of $C$ into $(0, 1)$ using the same method.

[8]The original range of $T(\mathbf{x})$ should be $(-\infty, +\infty)$. However, such range can cause stability problems for training. We adopt a standard trick, i.e., adding a small value to the log function, e.g., $\log(C(\mathbf{x}) + 1e^{-8})$, to change the range of $T(\mathbf{x})$ to $(-8, 8)$.

## D    MORE EXPERIMENTS AND ANALYSIS

In the paper we showed that PAN gets significant improvement comparing with state-of-the-art baselines. Here we give more detailed analysis of PAN in terms of dealing with varied positive ratio and the selection of hyper-parameter $\lambda$ in Eq. 3 (also 4).

### D.1    VARYING KNOWN POSITIVE RATIO

In this section, we analyse the performance of PAN when dealing with varied ratios of known positive examples. We vary the ratio of known positive examples from 5% to 30%, and show the accuracy and F-score of PAN and baseline on MNIST and CIFAR10 datasets. To verify the ability of PAN working in extreme conditions, we tested PAN on the situation with only 1% positive examples. The results are reported in Table 4 and 5.

Table 4: Varying the ration of known positive data on MNIST.

| Model | MNIST - results given as F / Acc | | | | |
|---|---|---|---|---|---|
| | 1% | 5% | 10% | 20% | 30% |
| NNPU | 88.34/88.51 | 93.96/94.09 | 95.60/96.51 | 96.89/96.96 | 97.51/ 97.57 |
| PAN | 90.45/90.30 | 95.27 /95.36 | 96.51/96.42 | 97.38/97.43 | 97.90/97.95 |

Table 5: Varying the ration of known positive data on CIFAR10.

| Model | CIFAR10 - results given as F / Acc | | | | |
|---|---|---|---|---|---|
| | 1% | 5% | 10% | 20% | 30% |
| NNPU | 81.41/84.22 | 86.09/88.84 | 87.84/90.14 | 89.05 / 91.04 | 90.01/91.66 |
| PAN | 82.70/86.10 | 87.22/89.70 | 88.37/90.77 | 89.74/91.85 | 90.65 / 92.49 |

From Table 4 and 5, we can see that PAN can do well with different proportions of known positive data and with extremely few known positive examples. We note that that the margin between PAN and NNPU goes large with the decrease of the ratio of known positive examples, which indicates that PAN is more effective than NNPU. The margin is smaller when the known positive ratio enlarges, that is because if we have enough positive data, the limitation of getting better results is no longer the PU learning method, but the performance of the classifier.

### D.2    HYPER-PARAMETER SELECTION

$\lambda$ is the hyper-parameter that balances the KL-divergences. Here, we show that $\lambda$ should be a small value but it is not too sensitive when it is around 0.0001. In our case, we set it to 0.0001.

Table 6: Sensitivity of $\lambda$ on MNIST.

| Model | Varying $\lambda$ - results are Acc | | | | |
|---|---|---|---|---|---|
| | 0.01 | 0.001 | 0.0001 | 0.00001 | 0.000001 |
| PAN | 82.70 | 88.90 | 90.30 | 88.23 | 86.44 |

### D.3    ABLATION STUDY FOR TERM MARKED BY III IN EQ. 3

Table 7: Accuracy (%) on different datasets for PAN with and without term III.

| Model | YELP | RT | IMDB | 20News | MNIST | CIFAR10 |
|---|---|---|---|---|---|---|
| PAN without term III | 80.67 | 60.32 | 78.45 | 72.63 | 96.30 | 89.38 |
| PAN full model | 83.56 | 64.10 | 78.84 | 81.00 | 96.42 | 89.70 |

Table 8: F-score comparison on different datasets with and without term III.

| Model | YELP | RT | IMDB | 20News | MNIST | CIFAR10 |
|---|---|---|---|---|---|---|
| *PAN without term III* | 81.86 | 66.28 | 78.52 | 73.59 | 96.27 | 86.68 |
| *PAN full model* | 83.45 | 66.58 | 77.10 | 81.06 | 96.51 | 87.22 |

Table 7 and 8 show PAN's ablation results in accuracy and F-score with or without term III respectively. From the two tables, we can see that adding term III indeed improves the performance of PAN on 5 out of 6 datasets.

