# OpenReview forum: "Learning from Positive and Unlabeled Data  with Adversarial Training"
_ICLR.cc/2020/Conference — Reject_

### Official Review · AnonReviewer3 · 2019-10-23
**Official Blind Review #3**

**Rating:** 3

**Review:**

<Paper summary>
The authors proposed a novel method for positive-unlabeled learning. In the proposed method, adversarial training is adopted to extract positive samples from unlabeled data. In the experiments, the proposed method achieves better performance compared with state-of-the-art methods.

<Review summary>
Although the idea to utilize adversarial training for PU learning is interesting, the proposed method is not sufficiently validated in theory. In addition, the manuscript is hard to follow due to confusing notations and lack of figures. I vote for rejection.

<Details>
* Strength
 + The main idea is simple and interesting.
 + The proposed method performs well in the experiments.

* Weakness and concerns
 - Confusing notations and lack of figures.
  -- Lack of mathematical definition of C and D.
  -- The argument of P^p and that of P^u are different (x^p and x^u), which implies that those distributions are defined at different space (but actually same).
  -- Shared index ``i" for positive and unlabeled data in Eq. (3).
  -- The notation with ``hat" often imply the empirically estimated (or approximated) value in the field of ML.
  -- No figures about the proposed method. Specifically, it is hard to understand the relationship between C and D.

 - Since Eq. (3) looks totally different from Eq. (2), why Eq. (3) is reasonable remains unclear.
  -- About I: first, P^{pu} cannot be calculated, because it requires unavailable labels of x^u. If you treat unlabeled data as negative, it should not be called ``ground-truth," and the term I cannot help D correctly recognize positive samples. Second, the positive samples are almost ignored in this term, because the number of positive data should be substantially small in a common setting of PU learning.
  -- About II: the authors explain the role of this term by min-max game between C and D during optimization, but the most important point here is what will happen when we obtain the optimal C and D after the optimization. What property or behavior do the optimal C and D have?

 - What do the authors want to claim with Proposition 1? The right-hand side of Eq. (5) cannot be easily calculated due to the density ratio between P^p and P^u. There is no explanation about what f and eps mean. What ``optimal" means is also ambiguous.


* Minor concerns that do not have an impact on the score
 - Although the problem setting is quite different, the idea of this paper is partially similar to the importance weighting technique adopted in some recent domain adaptation methods [R1, R2]. Do you have any comment on that?

[R1] ``Reweighted adversarial adaptation network for unsupervised domain adaptation," CVPR2018
[R2] ``Importance weighted adversarial nets for partial domain adaptation," CVPR2018



**Experience Assessment:**

I have read many papers in this area.

**Review Assessment: Checking Correctness Of Derivations And Theory:**

I assessed the sensibility of the derivations and theory.

**Review Assessment: Checking Correctness Of Experiments:**

I assessed the sensibility of the experiments.

**Review Assessment: Thoroughness In Paper Reading:**

I read the paper at least twice and used my best judgement in assessing the paper.

---

> ### Author Response · Authors · 2019-11-12
> **Thank you very much for your helpful comments. (PART 1)**
>
> Thank you very much for your helpful comments. We have addressed your concerns in the revised paper (uploaded). Below are our answers to your questions.
>
> Re: Confusing notations and lack of figures.
> 1.	Lack of mathematical definition of C and D:
>
> Response> Both C and D are neural networks and can be any of existing classification models. The exact architectures used in the paper are given in the experiment section (under Training Details) because C and D have different architectures for text and image. We also added this in the introduction section of the revised paper.
>
> 2.	The argument of P^p and that of P^u are different (x^p and x^u), which implies that those distributions are defined at different space (but actually same).
>
> Response> Different notations are used as we want to distinguish where the data come from. In SCAR (Selected Completely At Random) PU learning setting, positive data are randomly selected from the positive population. After that both positive and unlabeled sets are fixed before training. In this case, we use different notations because the unlabeled set is viewed as a whole and thus has a different distribution than the positive set because the unlabeled set also includes hidden negative data.
>
> 3.	Response> For the notation issues, we have addressed them in the revised version. We also added a figure to explain our method. We hope it is clear now.
>
> Re: -- About I: first, P^{pu} cannot be calculated, because it requires unavailable labels of x^u. If you treat unlabeled data as negative, it should not be called ``ground-truth," and the term I cannot help D correctly recognize positive samples. Second, the positive samples are almost ignored in this term, because the number of positive data should be substantially small in a common setting of PU learning.
>
> Response> Yes, we treat the unlabeled data as negative. So it can be calculated. Yes, treating unlabeled data as negative may not let D correctly recognize positive examples and thus decrease the performance of D. However, (1) we don’t need a strong D in the min-max game (see [1]), and (2) in the original GAN, the generated samples (or examples) also include both positive and negative data, i.e., the generated samples in GAN are basically like our unlabeled set. Term I aims to endow D with the ability to rank positives at the top. With this ability, the min-max game can work well iteratively.
>
> The positive samples will not be ignored in this term as we balanced the ratio of positive data and unlabeled data in each mini-batch in training. Thus, the impact of positive and unlabeled data is balanced. Sorry, we did not make this clear in the paper. It is added now.
>
> [1] Dai, Zihang, Zhilin Yang, Fan Yang, William W. Cohen, and Ruslan R. Salakhutdinov. "Good semi-supervised learning that requires a bad gan." In Advances in neural information processing systems, pp. 6510-6520. 2017.
>
> Re: “About II: the authors explain the role of this term by min-max game between C and D during optimization, but the most important point here is what will happen when we obtain the optimal C and D after the optimization. What property or behavior do the optimal C and D have?” and “- What do the authors want to claim with Proposition 1? The right-hand side of Eq. (5) cannot be easily calculated due to the density ratio between P^p and P^u. There is no explanation about what f and eps mean. What ``optimal" means is also ambiguous.” :
>
> Response> Intuitively, the behaviors of optimal C and D are that C gives the same prediction as D while D cannot move away from C, which means D also give high scores to examples that get high scores from C.
>
> We use Proposition 1 to decide the decision surface of our model. However, due to the complexity of the PU learning setting, f() is very complex (but it can be computed). Thus, we showed its properties, which we believe is sufficient. Please see Appendix C. We added explanations of the behaviors of optimal C and D, and moved Proposition 1 to Appendix C.
>
> Hope our responses are clear. If you have any additional questions, please let us know. We will address or clarify them.

---

> > ### Comment · AnonReviewer3 · 2019-11-13
> > **Thank you for your response**
> >
> > Thank the authors for the response.
> >
> > 1. My comment is not about how C and D are implemented but about their mathematical definitions. Specifically, clarifying the input and output space of the function is important. The authors use C(.) for a vector-to-scalar mapping, though C_i is for a vector-to-vector mapping.
> >
> > 3. My major concern is "whether or not we can obtain the optimal (unbiased) classifier by the optimization in Eq. (3)?". It is obvious in case of Eq. (2), because D performs worst if C perfectly extracts the positive samples from unlabaled data. On the other hand, it is not clear in case of Eq. (3). When we only consider term I, D is biased since the unlabaled data contain the positive samples. I imagine that term II and III reduce this bias to obtain the unbiased classifier C, but it is not clearly shown. (While I understand the authors' intention described in section 4.1, it is not supported well in theory.) In addition, tuning lambda seems to play almost same role with the class-prior estimation, if the above intuition is correct.

---

> > > ### Author Response · Authors · 2019-11-15
> > > **Thank you very much for your new comments.**
> > >
> > > Thank you very much for your new comments.
> > >
> > > Re: My comment is not about how C and D are implemented but about their mathematical definitions. Specifically, clarifying the input and output space of the function is important. The authors use C(.) for a vector-to-scalar mapping, though C_i is for a vector-to-vector mapping.
> > >
> > > Response> Yes, C(.) is a vector-to-scalar mapping, C_i denotes the probability distribution of the data point x_i being positive and negative, which is a vector-to-2dimension mapping. The 2 dimensions are the outcome positive and negative probabilities, i.e., (C(x_i), 1 – C(x_i)).
> > >
> > > Re: My major concern is "whether or not we can obtain the optimal (unbiased) classifier by the optimization in Eq. (3)?". It is obvious in case of Eq. (2), because D(.) performs worst if C(.) perfectly extracts the positive samples from unlabeled data. On the other hand, it is not clear in case of Eq. (3). When we only consider term I, D(.) is biased since the unlabeled data contain the positive samples. I imagine that term II and III reduce this bias to obtain the unbiased classifier C(.), but it is not clearly shown. (While I understand the authors' intention described in section 4.1, it is not supported well in theory.) In addition, tuning tuning lambda seems to play almost same role with the class-prior estimation, if the above intuition is correct.
> > >
> > > Response> Our method PAN (Eq. (3) or (4)) basically follows the similar idea to that of Eq. (2). In Eq. (2), D performs worst if C(.) perfectly extracts the positive samples from the unlabeled set, which is correct as D(.) is unable to classify/separate the given positive data and the possible positives x’ extracted by C(.). In the case of PAN, that also means D(.) gives high positive probability scores to x’ like C(.). Thus, the final training result is that D(.) and C(.) give similar predictions, or D(.) cannot move away from C(.) (meaning D(.) also gives high scores to the examples that get high scores from C(.)).
> > >
> > > We agree that Eq. (3) is harder to understand as it is different from GAN and also because for KL-divergence, the probabilities of a distribution can go up or down in order to match another distribution, which makes it more difficult to explain as there are many cases. Yes, your intuition is correct. Terms II and III try to correct the bias of D(.) in term I of Eq. (3). But let us see the idea using Eq. (4), which is derived from Eq. (3) for training and it is much clearer than Eq. (3). Note that the bias in term I in Eq (3) will result in high precision and low recall for the positive class. Now back to Eq (4) and let us imagine that most examples are regarded as negatives by C(.) (low recall). Then, from Eq. (4), we can see the value of term V is below zero. When optimizing D(.), term VI will push D(.) up for these examples, and thus the bias is reduced and the low recall problem is mitigated as in the next optimization iteration, C(.) for the examples will also go up following D(.). We have added more explanation in the paper using Eq. (4) in Appendix C.
> > >
> > > Regarding lambda, in our experiments it is fixed to 0.0001 for all experiments. It can have some indirect effect of correcting the bias but we believe the main effect is from above.
> > >
> > > Hope our explanation is clear. If you have any additional questions, please let us know. We will address or clarify them quickly.

---

> ### Author Response · Authors · 2019-11-12
> **Thank you very much for your helpful comments. (PART 2)**
>
> Thank you very much for your helpful comments. We have addressed your concerns in the revised paper (uploaded). Below are our answers to your questions.
>
> Re: “Although the problem setting is quite different, the idea of this paper is partially similar to the importance weighting technique adopted in some recent domain adaptation methods [R1, R2]. Do you have any comment on that?”
>
> Response> Thanks for pointing this out and the two relevant papers. We read the two papers and have cited them and compared them with our work. Our a-GAN method has some similarity with the weighted adversarial nets (WAN), but our PAN differs significantly from WAN (as PAN differs significantly from a-GAN). That is because although WAN weighted D by w(z) but the adversarial training procedure is the same as the original GAN (similar to our a-GAN). The examples generated by G are fed into D for discrimination. A-GAN uses the same strategy, which did not work well in our case. That is why we designed a new formulation in PAN, which uses KL-divergence and the three terms in Eq. (3) to solve the problem as we discussed in Section 4.
>
> Hope our responses are clear. If you have any additional questions, please let us know. We will address or clarify them.

---

### Official Review · AnonReviewer1 · 2019-10-23
**Official Blind Review #1**

**Rating:** 3

**Review:**

This paper considers the problem of learning a binary classifier from only positive and unlabeled data (PU learning), where they develop a Predictive Adversarial Networks (PAN) method by using the GAN-like network architecture with a KL-divergence based objective function. Experiments and comparisons with SOTA are provided.

Pros:
Their idea of making an adaption to GAN architecture by replacing the generator by a classifier to select P from U and using the discriminator to distinguish whether the selected data is from P or U for PU learning is interesting, and benefits from not relying on the class prior estimation.

Question:
The authors claim that Eq. (2) cannot be trained in an end-to-end fashion directly, this statement may need some modification since there are some existing works replacing c(x) by a score function or some other continuous function and then this direct adaptation can be trained, for example, see Eq. (5) in “Discriminative adversarial networks for positive-unlabeled learning. arXiv:1906.00642, 2019”. Can any explanation be given on this?

Remarks:
The clarity of the paper could be improved in multiple places. For example, the data generation processes can be mathematically defined in the problem setting part, now it is quite confusing to me. And more details on experimental protocol may be needed: e.g. what kind of hyperparameter tuning was done?

In general, the paper proposed an interesting GAN-like network architecture to learn from PU data, but some unclear parts need to be improved.

**Experience Assessment:**

I have published one or two papers in this area.

**Review Assessment: Checking Correctness Of Derivations And Theory:**

I assessed the sensibility of the derivations and theory.

**Review Assessment: Checking Correctness Of Experiments:**

I assessed the sensibility of the experiments.

**Review Assessment: Thoroughness In Paper Reading:**

I read the paper at least twice and used my best judgement in assessing the paper.

---

> ### Author Response · Authors · 2019-11-12
> **Thank you very much for your helpful comments.**
>
> Thank you very much for your helpful comments. We have addressed your concerns and improved the clarity of the paper and uploaded it.
>
> Re: The authors claim that Eq. (2) cannot be trained in an end-to-end fashion directly, this statement may need some modification since there are some existing works replacing c(x) by a score function or some other continuous function and then this direct adaptation can be trained, for example, see Eq. (5) in “Discriminative adversarial networks for positive-unlabeled learning. arXiv:1906.00642, 2019”. Can any explanation be given on this?
>
> Response> We are wondering where we made that claim. We did not make that claim in the submitted version. Could you please check again? If you find that claim, please let us know the location and we will definitely revise it. Actually, our a-GAN is trained in an end-to-end fashion. The recent arXiv paper (which should be done at about the same time as our paper) that you mentioned is similar to a-GAN, but our PAN differs from it significantly. We have cited and discussed it in the revised version.
>
> For other questions:
> Response> Thank you for your suggestions to improve the clarity of the paper. We have revised it and make things clearer. About hyperparameter tuning, we gave an analysis in Appendix D.2. Could you refer to that for more details and let us know whether it is satisfactory.
>
> If you have any additional questions, please let us know. We will address or clarify them.

---

### Official Review · AnonReviewer2 · 2019-10-25
**Official Blind Review #2**

**Rating:** 6

**Review:**

The paper proposed an interesting idea of using two adversarial classifiers for PU learning. The first classifier tries to introduce samples from unlabeled data that are similar to the existing labeled positive data, and the second one tries to detect if a sample has a ground truth label or drawn from unlabeled data by the first classifier (fake). The idea of the paper is interesting; it is well-motivated and well supported with a range of experiments from NLP and computer vision tasks. The paper is a good read but required a pass of proofreading; some rearrangement of the concepts (for example, in second paragraph C(.) and D(.) is used, but they are introduced properly in section 4. Also, the paper could use some clarifications.

* How the proposed method handles unlabeled positive samples that have a different distribution from P or has a similar distribution with some of the negative samples that might exist in unlabeled samples.

* The experiment section could have enjoyed from an ablation study in which a system that only implements terms I and II from Eq(3). The authors mentioned that such an objective function is asymmetric but didn't explore the implications of such an objective function in the empirical experiments.

* PGAN's results are not compared in the fair condition since the PU version of CIFAR 10 is different from PGAN's version.

* Using MLP classifier for the text classification (e.g., for YELP) makes a very weak baseline for the system. Also, training the word embedding by the system itself is unrealistic. Therefore, the sentence might need to be rewritten.

* Some readability issues:
(i) C(.) and D(.) needs an introduction in the section "I Introduction" before their usage.
(ii) The idea could be illustrated easily. Such a figure significantly improves the readability of the system.
(iii) be careful with the use of \cite{} and \citep{} interchangeably ("{Liu et al., 2003; Shi et al., 2018" -> Liu et al. (2003) and Shi et al. (2018) ...,
(iv) The first paragraph of section 2 should be split into two from this phrase "None of these works..."
(v) Please rewrite the latter half of paragraph 2 in section 2. Also, please rewrite the beginning sentences of section 4.1 and the final paragraph of section 3.
(vi) right after equation (2), please change x_s to \mathbf(x)^s for the consistency of your formulation.
(vii) favorible -> favorable, radio (in section 5.1) -> ratio,
(viii) please add a reference for this statement. "This is one of the best architectures for CIFAR10."

**Experience Assessment:**

I have published one or two papers in this area.

**Review Assessment: Checking Correctness Of Derivations And Theory:**

I assessed the sensibility of the derivations and theory.

**Review Assessment: Checking Correctness Of Experiments:**

I carefully checked the experiments.

**Review Assessment: Thoroughness In Paper Reading:**

I read the paper thoroughly.

---

> ### Author Response · Authors · 2019-11-12
> **Thank you very much for your helpful comments.**
>
> Thank you very much for your helpful comments. We have addressed your concerns in the revised paper, which has been uploaded.
>
> Re: * How the proposed method handles unlabeled positive samples that have a different distribution from P or has a similar distribution with some of the negative samples that might exist in unlabeled samples.
>
> Response> In this paper, we do not consider this case. We assume the examples in the positive set are sampled randomly from the positive population. The problem that you mentioned is very challenging. We will consider it in our future work.
>
> Re: * The experiment section could have enjoyed from an ablation study in which a system that only implements terms I and II from Eq(3). The authors mentioned that such an objective function is asymmetric but didn't explore the implications of such an objective function in the empirical experiments.
>
> Response> We actually did the experiments before and the results were not good for YELP, 20news and RT. That’s why we employed the third term. We have redone the experiments and added the results in Appendix (D.3).
>
> Re: * PGAN's results are not compared in the fair condition since the PU version of CIFAR 10 is different from PGAN's version.
>
> Response> Thanks for pointing this out. We apologize. We could not and still cannot find the code of PGAN online and we thus are unable to run the experiments. However, we ran our method PAN in PGAN's setting on CIFAR10 and we get much better results, 91.16 for accuracy and 89.13 for F-score while the F-score of PGAN is 76 as reported in the PGAN paper. This much better result of PAN is mainly because PGAN’s setting uses much more positive data than our setting in the paper. We believe that less positive data is more realistic in practice. Thus, as we cannot compare with PGAN in our setting, we believe it is better to remove that specific result (F-score=76) that we quoted in the submitted version earlier from the revised paper. However, we still cited and compared with it in the related work section. If you know there is a version of the PGAN code online, please let us know and we will run it immediately.
>
> Re: * Using MLP classifier for the text classification (e.g., for YELP) makes a very weak baseline for the system. Also, training the word embedding by the system itself is unrealistic. Therefore, the sentence might need to be rewritten.
>
> Response> Thanks. We used pre-trained word embeddings learned by the skip gram method of word2vec on the corresponding datasets. Perhaps there is a misunderstanding about classifier. The classifier is a 2-layer convolutional network (CNN), with 5 * 100 and 3 * 100 convolutions for layers 1 and 2 respectively, and 100 filters for each layer. An MLP layer follows to map the output features to the final decision scores. Only the MNIST dataset uses MLP only.
>
> **Thanks for pointing out some readability issues. We have revised the paper and uploaded the new version. If you have any more questions, please let us know. We will address them and make everything clear.

---

### Decision · Program_Chairs · 2019-12-19

**Decision:**

Reject

**Comment:**

Thanks for your feedback to the reviewers, which helped us a lot to better understand your paper.
Through the discussion, the overall evaluation of this paper was significantly improved.
However, given the very high competition at ICLR2020, this submission is still below the bar unfortunately.
We hope that the discussion with the reviewers will help you improve your paper for potential future publication.